# Wastewater Reuse to Mitigate the Risk of Water Shortages: An Integrated Investment Appraisal



Foroogh Nazari Chamaki [1], Hatice Jenkins [1], Majid Hashemipour [2] and Glenn P. Jenkins [3],*

1. Department of Banking and Finance, Eastern Mediterranean University, Famagusta 99628, North Cyprus, Turkey
2. Faculty of Engineering, Cyprus International University, Nicosia 99258, North Cyprus, Turkey
3. Department of Economics, Queen's University, Kingston, ON K7L 3N6, Canada
* Correspondence: jenkinsg@queensu.ca

**Abstract:** This paper evaluates the financial and economic costs of reusing wastewater with reverse osmosis (R.O.) purification systems to mitigate the risks of near potable quality water shortages in an urban water system. A distributional analysis is also undertaken to identify those who bear the externalities of the system. A rich data set is available to conduct an ex-post analysis of such a system operating in Cyprus for several years. The levelized financial cost of the R.O. system if it operates at a 75% utilization rate is USD $1.18/m^3$, while the levelized economic cost that includes all the externality impacts is USD $1.20/m^3$. However, the closeness of these two values hides a large set of externalities that affect different groups in society in disparate ways. The analysis shows that reusing wastewater in conjunction with a system of R.O. is a very effective way to mitigate the risks of water shortages in a more extensive water system. It also highlights the importance of the nature of the electricity system that generates the electricity to power the R.O. plant in determining the ultimate economic cost of reusing wastewater.

**Keywords:** reused wastewater; reverse osmosis; water shortage risk; levelized cost; economic cost; emission cost; environmental externalities; distributive analysis





## 1. Introduction

Many countries, especially in the Middle East and North Africa, are facing increased challenges in meeting the growing water demand in their urban areas. The risk of water shortages is further increased due to the prospect of growing fluctuations in precipitation because of climate change that will impact the available water supply.

In the water sector, the planning and management issues are evolving to place increasing emphasis on the mitigation of risks brought about by climate change. The challenge now is to plan and build water supply systems cost-effectively to mitigate the risk of shortages due to abnormal fluctuations in the demand for and supply of water. Water storage of seasonal rainfall through dams and the development of deep wells has been the traditional method of supplying the needs of a community over time. However, the marginal cost of supplying water from these traditional technologies is increasing due to the distance of piping, the increasing costs of reservoirs, and the scarcity of conventional water resources.

The approach to mitigating electricity outages in the electricity sector has valuable lessons for countries seeking to manage the risk of water shortages. To meet consumers' instantaneous demand for electricity, the electric utility must invest in the generation capacity of technologies that can quickly respond to close any gap between the amount of electricity demanded and the current supply [1,2]. The demand for electricity by consumers and the supply of electricity generated may fluctuate daily and seasonally. Single cycle electricity generation plants typically make up a large proportion of such reserve generation

capacity. Much of this reserve electricity generation capacity will operate for a relatively small proportion of the time over a year.

To improve the reliability of water services, a similar approach is suggested in this article for some countries. Here, an evaluation is made of the feasibility of building a rapid response water supply by investing in reverse osmosis (R.O.) plants that utilize treated urban wastewater.

Treated wastewater supplies are available close to urban areas, and the quantities of water are pretty stable as they are determined mainly by the number of residents in the areas. The water supply produced by the reverse osmosis system that cleans this urban wastewater can be quickly adjusted to meet any supply gaps between the quantity of water demand and the quantities supplied from conventional sources.

For communities close to the sea, desalination of seawater or brackish water is one possible source for use as an input into an R.O. system. This water source is often expensive and is only available to communities with a seacoast. Another source of potable water that is close to urban areas, and is very reliable in supply, is urban wastewater purified through advanced R.O. treatments.

The focus of this study is to estimate the willingness to pay for risk mitigation based on the costs that water suppliers in a community in North Cyprus incur to reduce the risk of water shortages. The data for this study is obtained from an ex-post analysis of the costs of the coping strategy for maintaining water service reliability for this community. The location is at the approximate center of the island of Cyprus. The willingness to pay is estimated here by the levelized financial and economic cost of operating an R.O. plant treating wastewater from the largest urban area on the island. It is designed to supply water at different capacity utilization levels during the year. It can respond quickly to meet any gap that might arise between the various areas demanding water and the traditional sources of supply of water to this community.

Previous studies have analyzed how households attempt to maximize their utility given their constraints of both monetary income and time in their selection of water sources to meet their demand for a set of qualities and quantities of water [3]. Our research focuses on how a specific community has selected diverse water sources to meet fluctuating demand for water of different qualities. The lessons from this analysis will apply to other communities that face similar circumstances.

The demands for water in this community are very diverse. They come from a major university, primary and secondary schools, a large dairy farm, vineyards, greenhouses, several manufacturing enterprises, and many households. Although Cyprus is surrounded by seawater, the costs of desalination plus the transport of the water from the coast to this community would be very high.

The primary water sources for this community besides the R.O. of wastewater are wells and the water supplied by various municipal governments. The minimum quality of water is quite different for the irrigation of tree crops compared with its use by households and educational institutions. On average, the community consumes approximately 3000 m$^3$ of water daily. Around 1700 m$^3$ of the water consumed per day is provided by the municipal water utilities and 700 m$^3$ from wells. The remainder is supplied by the R.O. plant using treated wastewater as input. The plant operates daily over the full range of its capacities, from operating for only a few hours a day to operating at full capacity. On average, over a year, the R.O. plant operates at about 75% of its full capacity.

Due to Cyprus's wide range of seasonal temperatures, its water demands fluctuate widely. The student population size also changes significantly between the months of the academic year and the holiday periods. These factors add to the variability of the demand for water over time. At the same time, the supply of water from the government is unreliable. Water rationing by municipalities to their consumers has been a common practice for decades.

Furthermore, the low quality of the water from the wells is often unsuitable for some purposes. Consequently, the gap between the water demanded in this community and the

available supply from the traditional municipal and wells sources is highly variable. The social costs of temporary shortages are also very substantial. The option of diverting water from relatively low-value uses, such as crop irrigation, to high-value uses is minimal, as crop irrigation in this community is very limited. For the community to be sustainable, it needs to have a cost-effective system for managing the sources of supply of water so that significant water shortages do not occur.

In 2015 a relatively small R.O. plant was built to desalinate brackish water from wells. It was designed to produce approximately 1000 m$^3$ per day. The low quality of the input source made the water supplied by the plant expensive and not operational. Consequently, it was decided to switch to purchasing treated wastewater from the Nicosia Wastewater Treatment Plant (NWTP) as the feed-in water for the R.O. plant. Since late 2019 the community has been purchasing water from NWTP and employing R.O. treatment to produce water of sufficient quality to meet its demands on an as-needed basis. The plant continually replenishes a water storage tank connected to the R.O. plant. This stored water can be delivered immediately by pipelines or tanker trucks wherever needed. This way, the risk and costs of water shortages for the entire community are mitigated. In practice, reusing wastewater via the R.O. process has been an excellent complement to the other sources of supply in risk mitigation. Several studies, for example, by Arborea et al. [4], have focused on the value of the reuse of treated wastewater primarily for irrigation. However, in this situation, the need is for a near-potable water supply system with high reliability. The research question is, what is the cost of maintaining reliability in the overall water supply system using this technology?

Because this plant has been operating for the past three years, a rich set of information on production and costs is available to estimate the marginal cost of maintaining reliability using the R.O. wastewater treatment as an element in the overall water supply system. The Levent Group in North Cyprus has provided all the project-specific data to the authors. In this study, we conduct an integrated investment appraisal to determine its financial and economic costs as a risk-mitigation method.

## 2. Literature Review

The risk associated with the volatility of the gap between the demand and supply of water in urban communities is expected to increase over time [5]. Population growth, urbanization, and industrialization will increase water demand [6]. The risk of water shortages, given the impacts of climate change, is almost certain to increase over time. Hence, to maintain the sustainability of urban water supply systems, the investment planning and appraisal functions need to mitigate the risk of water shortages cost-effectively.

The reuse of wastewater has been increasingly discussed as an essential new resource to be used to address this challenge. Its great advantage is that it produces a highly reliable water source wherever households and industries consume water. While limited in volume, R.O. treatment plants can quickly convert this into high-quality water.

In conjunction with short-term storage, water produced by such a system can create a negative correlation between this supply and other available supplies of water to the system. Alternatively, a positive correlation can be created between the supply of water from this source and the gap between the demand for water and the supply of water from other sources. The result will be a lower level of overall risk and cost imposed by water scarcity on water consumers. The R.O. plants using wastewater as feedstock can play the same role in mitigating shortages as a peaking electricity generator in building reliability into an electricity service. In addition to fostering sustainable growth and assisting the shift to a circular economy, continuously improving and reasonably priced advanced treatment technologies offer reused wastewater as a dispatchable supply of clean water.

The technique of reusing treated municipality wastewater through R.O. membrane technology has been used in many countries around the globe. In the late 1800s and early 1900s, the development of relatively advanced treatment systems resulted in the evolution of wastewater reuse [7]. Most significantly, Israel, the world leader in water reuse,

reclaims 400 million m$^3$ per year of its domestic advanced treated wastewater and uses it for agricultural irrigation [8]. The first reclamation facility in Windhoek, the capital and largest city of Namibia, one of the driest nations in Africa, began operating in 1968 with a capacity of 4800 m$^3$ per day. Direct potable water reclamation from wastewater is produced at the Goreangab Reclamation Plant [9–11]. Due to severe water shortages, a direct wastewater reclamation plant for producing drinking water was constructed in Beaufort West, South Africa, at the end of 2010. It has a daily production capacity of 2300 m$^3$ [12].

The first system for replenishing potable groundwater using wastewater in the United States was established in Fresno, California, in the 1920s [13]. In the 1960s, in Irvine Ranch, California, water reuse was very effectively implemented on a large scale for a whole community. An integrated dual-pipe system with almost 450 miles is the baseline in the Irvine project, ensuring that drinking water and recycled water are distributed separately for the entire city.

Since 2010 the number of potable reuse projects has quadrupled in the United States. Such projects are now widely acknowledged as a critical part of urban water security portfolios [14]. In 2008 the Groundwater Replenishment System (GWRS), the largest advanced water purification facility in Orange County, California, produced 265,000 m$^3$ per day of recycled water of drinking quality that exceeds state and federal potable water standards. The GWRS cost an estimated USD 481 million to design, expand, and construct [14–16]. Spain has the highest rate of wastewater reuse in Europe, while China and Mexico topped the list of nations that reuse wastewater globally [17].

Reusing wastewater for irrigation has been a critical element in meeting the demand for water in the Republic of Cyprus [18–20]. The availability and reuse of wastewater for irrigation have been critically important as it has allowed the water authorities to ease the rationing of water for irrigation. This substitution of treated wastewater for conventional sources has allowed the water released from agricultural irrigation to be diverted to residential and urban water consumers. Additionally, the investment in a substantial capacity of desalination plants has been a key element in managing water shortages in years of drought.

A vital issue to consider when reusing wastewater is the water quality required to address specific consumers' needs. Different levels of treatment are required to supply the quality of water demanded by different consumers [21]. The desired quality of water is addressed by using different technological processes to purify the water. In the E.U., the stringent water quality conditions enable the acceptance of wastewater reuse to meet the demand of various sectors [22].

## 3. North Cyprus Water Supply Developments

Cyprus is the third largest island in the Mediterranean Sea. It is a semi-arid region. It has a surface area of approximately 9251 km$^2$ surface area [23] and a population of approximately 1,200,000 [24]. North Cyprus constitutes roughly 30% of the island of Cyprus, with a population of about 370,000.

Until 2015, when a water pipeline was built between North Cyprus and Turkey, the island's largest water source was groundwater and surface water. From the 1960s, consecutive droughts, overuse of water, and increasing demand for water for agriculture caused a fall in water levels in aquifers on the island to below their capacity and the depletion of groundwater resources in quantity and quality. Excessive water extraction led to the penetration of seawater into the aquifers, damaging water quality and making it unpotable and unusable for some types of agriculture. The existing water sources on the island could not meet the water demand.

Moreover, due to the number of dams built in the south of the island, insufficient water reaches the dams and aquifers in North Cyprus, mainly the acquirers of Kyrenia and Guzelyurt.

North Cyprus has an annual water potential of 117.5 million m$^3$/year, including 27 million m$^3$/year from rivers, 1.4 million m$^3$/year from springs, and 89.1 million m$^3$/year

from groundwater. The region's primary water source is the Troodos mountains. While only 74.1 million m$^3$/year should have been withdrawn annually from its aquifers, 28.9 million m$^3$/year of excess water is harvested, causing their depletion [25].

Following attempts to bring water from Turkey to North Cyprus using balloons and tankers [26], it was finally decided to bring water through an undersea pipeline from Turkey. This project was designed in 1996 by the Turkish Republic General Directorate of Mineral Research and Exploration (M.T.A.). About 75 million m$^3$/year of water can be transmitted through this project from Turkey to North Cyprus. This 80 km pipeline was completed in 2015 [27].

Since 2015, freshwater coming directly to North Cyprus through the undersea pipeline has partially resolved the water shortage. With gravity, this water is transferred from the Alakopru Dam reservoir in Turkey to the Gecitkoy Dam reservoir in Kyrenia, North Cyprus [28,29].

While the water coming to North Cyprus from Turkey has dramatically improved the water situation in terms of quantity and quality of water supply, it has not mitigated all the water availability risks. In 2020 there was a break in the pipeline, and for ten months, water was supplied to consumers from water stored at the Gecitkoy Dam reservoir. By the time the pipeline was repaired, the water supply from the Gecitkoy Dam reservoir was exhausted, and widespread rationing was implemented. Several municipalities had to revert to wells that supplied very low-quality water. Furthermore, because of the depreciated state of the local water distribution systems, periodically, water is not supplied continuously to the final consumers, even when there is water in the large reservoirs. This has led the community in this study to consider either desalinating seawater or employing advanced R.O. wastewater treatment as possible sources of high-quality water that can mitigate the risk of water shortages.

South Cyprus first introduced seawater desalination in 1997 to mitigate a severe water shortage. From its introduction to 2007, approximately 40% of the potable water supply was provided by desalination. The proportion of water coming from desalination peaked at 70% in 2009. Due to the increase in water supplied by dams, by 2013, the proportion of the water provided by desalination plants had fallen to 14% of total consumption [30]. Currently, approximately 80% of the capacity for desalination is on standby reserve capacity but can be brought into service quickly should the need arise.

As the municipal water distribution systems constrain the water supply to communities, the fluctuation of a community's demand and supply would generally require a substantial number of local reservoirs to respond to the seasonal fluctuations in the demand for and supply of water. One of the ways to ensure the water supply's security and reliability is to have the capacity to treat wastewater reuse to a high-quality level. The wastewater available as input to this R.O. system is always available, and the quality is much better than seawater or brackish water for treatment. The cost of enhanced wastewater treatment through R.O. will be lower than seawater desalination due to better feed-in water quality requiring lower electricity costs to purify it.

A further risk for consumers of the municipal water supply is price risk. In 2015, when the water first came to North Cyprus by pipeline, the price paid to Turkey for the water per m$^3$ was TL2.3 or USD 0.85 [31]. The price charged by the municipalities to consumers averaged about TL4.5/m$^3$ or USD 1.66/m$^3$ [32]. Seven years later, the government of Turkey still charges North Cyprus TL2.3/m$^3$. Using the exchange rate of September 2022, this is only USD 0.13/m$^3$. In September 2022, depending on the municipality, the water from Turkey was sold to consumers at a price ranging from TL6.5/m$^3$ to TL15.0/m$^3$, or from USD 0.36/m$^3$ to USD 0.82/m$^3$ [31]. Municipal water charges are politically determined in North Cyprus. Before the water came from Turkey, the municipalities relied on water sales as a significant source of funds for financing other activities. The water was mainly from municipal wells, where the pumping costs were approximately USD 0.18/m$^3$ [33]. However, municipalities sold this low-quality water to their water consumers at multiples of their costs, generating substantial profits. Given the severe financial stress many municipalities

are experiencing, increasing water charges would be a desirable policy in the current inflationary environment. Hence, a significant consumer of water, such as the community in this study, needs to have a risk-mitigating strategy that provides some certainty as to the maximum cost of supplying good-quality water.

*Availability of Treated Wastewater in North Cyprus*

Wastewater is increasingly being recognized as a potential 'new' source of clean water for near potable and potable uses. There are four wastewater treatment plants in North Cyprus: Nicosia, Gazimagusa, Girne, and Guzelyurt. NWTP, a bi-communal plant located in North Nicosia, is the largest. It is Europe's second-largest wastewater treatment plant that uses advanced membrane technology. This plant has been in operation since July 2013. This wastewater treatment plant produces a high-quality output with a ppm of (800), 1600 uS/cm conductivity, and a P.H. of 7.7.

On average, it supplies over 30,000 $m^3$ of treated wastewater daily. In other words, 11,000,000 $m^3$ of treated wastewater per year is available through this plant. The plant was designed to supply treated water and sludge for agricultural purposes. In the initial agreement, 30% of the wastewater and sludge would eventually be distributed to North Cyprus and 70% to South Cyprus [29,34].

In South Cyprus, treated wastewater is used extensively for agricultural irrigation. Over 70% of all treated wastewater is utilized in agriculture. In North Cyprus, only 1250 $m^3$ per day of treated wastewater from the Guzelyurt wastewater treatment plant is currently used for agricultural irrigation [35]. The rest of the treated water, except that used by this R.O. plant, is used to augment the depleted aquifer reserves.

## 4. Methodology

This study undertakes an analysis to estimate the financial and economic costs of the community's effort to increase the reliability of the water supply through wastewater treatment so that it can be used to augment the other sources of supply. The primary function of this water supply is to maintain the overall reliability of the water service. In this study, following the integrated investment appraisal approach of Jenkins, Kuo, and Harberger [36], the financial and economic costs of the investment and operation of the plant are quantified.

The financial analysis of this R.O. plant is carried out using a financial cash flow model of the costs of the system for enhancing the quality of treated wastewater. The analysis has been conducted for 20 years. Similarly, an economic resource flow model is developed to evaluate the economic costs of this system for the same duration. By comparing the financial cash flow costs with the economic resource outflow flows period by period, one can measure the distributional impacts of this facility. These impacts include the financial impacts on the private owners and the economic welfare losses arising from both local pollution and global costs inflicted by the increase in greenhouse gases from the electricity generation used for the operation of the R.O. plant.

The theoretical foundation for the integrated investment appraisal methodology is found in the seminal article by Arnold C. Harberger [37]. This methodology has been applied to analyze the distributional impacts of interventions in many different sectors. For example, it was applied in the analysis of the Yesilirmak Dam in North Cyprus [38] and the evaluation of wind farms in Ontario, Canada [39]

The financial costs of the supply from the R.O. system at different capacity utilization levels provide information on the levelized cost of supplying water from the plant to provide water system reliability.

The output water from the R.O. system has the following characteristics: ppm (40), Conductivity (50 uS/cm) with a P.H. of 6.5. Because of the high quality of water it produces, such a plant can provide reliability in terms of the quality and quantity of water. The economic analysis quantifies the economic resource costs incurred to produce this water from the perspective of the residents of North Cyprus, of all of Cyprus, and globally.

The economic costs must exclude several financial costs from the analysis, such as taxes and payments for the purchase of the wastewater from the municipality, as these are not net costs from the point of view of the economy. At the same time, they must include external costs, such as the opportunity cost of the wastewater used by the R.O. plant that would have been used to recharge the aquifer. The economic costs under some circumstances include the costs of the disposal of the effluent from the R.O. process that the plant owners do not bear. A major external economic cost arises because additional electricity is required to power the R.O. plant. In North Cyprus, electricity is generated primarily by thermal plants that use heavy fuel oil (H.F.O.). The operation of these unfiltered diesel generation plants creates significant pollution and damages people's health in Cyprus. It also increases greenhouse gas emissions. The effects of the environmental impacts are quantified by the North Cyprus residents, All Cyprus residents, and the global population. The economic costs will include all the items that are a cost to the residents of North Cyprus, all of Cyprus, and globally.

The initial focus of the analysis is to quantify the financial and economic levelized cost of the water output from this plant. A levelized cost analysis is carried out for 20 years. The amount of water is determined by the average hours the plant operates each day over its lifetime. The present value of the water produced over this period is estimated by discounting each annual quantity of water produced by the applicable financial or economic discount rates. The financial cost includes the capital costs as well as the operating costs of the plant over time. It also includes the cost of purchasing the feed-in wastewater and the pumping costs both into and out of the plant.

The following 12 equations define how each component of the subsequent financial, economic, and distribution analyses is defined and quantified. The table of parameters has been provided in Table 1.

Present value (P.V.) @ 8% of the water produced over the plant's lifetime:

$$\text{P.V.}_{\cdot t=0}^{\text{P.W.}} = \sum_{t} \text{Q.P.W.}_{\cdot t} * (1+r)^{-t} \tag{1}$$

Financial costs incurred in year $t$:

$$\text{C}_t^F = \text{P.M.X.}_{\cdot t}^F + \text{E.C.X.}_{\cdot t}^F + \text{ChCX}_t^F + \text{FO\&MX}_t^F + \text{ICAPEX}_t^F + \text{RCAPEX}_t^F \tag{2}$$

Present value of financial costs for plant:

$$\text{P.V.}_{\cdot t=0}^{\text{F.C.W.}} = \sum_{t=0}^{T=21} \text{C}_t^F * (1+r)^{-t} \tag{3}$$

Levelized financial cost per m$^3$ of pure water expressed in the price level of 2022 (Equation (3)/Equation (1)):

$$\text{L.C.}^F = \frac{\sum_{t=0}^{T=21} C_t^F * (1+r)^{-t}}{\sum_{t=0}^{T=21} \text{Q.P.W.}_{\cdot t} * (1+r)^{-t}} \tag{4}$$

The economic cost in year $t$ of plant:

$$\text{C}_t^E = \text{PM}_t^E + \text{E.C.}_{\cdot t}^E + \text{ChC}_t^E + \text{FO\&M}_t^E + \text{ICAPE}_t^E + \text{RCAPE}_t^E \tag{5}$$

$$\text{C}_t^{ENC} = \text{PM}_t^E + \text{E.C.}_{\cdot t}^E + \text{ChC}_t^E + \text{FO\&M}_t^E + \text{ICAPE}_t^E + \text{RCAPE}_t^E + \text{EC}_t^{NC} \tag{6}$$

$$\text{C}_t^{ECY} = \text{PM}_t^E + \text{E.C.}_{\cdot t}^E + \text{ChC}_t^E + \text{FO\&M}_t^E + \text{ICAPE}_t^E + \text{RCAPE}_t^E + \text{EC}_t^{CY} \tag{7}$$

$$\text{C}_t^{EG} = \text{PM}_t^E + \text{E.C.}_{\cdot t}^E + \text{ChC}_t^E + \text{FO\&M}_t^E + \text{ICAPE}_t^E + \text{RCAPE}_t^E + \text{EC}_t^{CY} + \text{EC}_t^{GHG} \tag{8}$$

Levelized economic costs per m$^3$ of pure water expressed in prices of 2022:

$$\text{L.C.}^E{}_{22} = \frac{\sum_{t=0}^{T=21} C_t^E * (1+r)^{-t}}{\sum_{t=0}^{T=21} \text{QPW}_t * (1+r)^{-t}} \tag{9}$$

$$\text{L.C.}^{ENC}{}_{22} = \frac{\sum_{t=0}^{T=21} C_t{}^{ENC} * (1+r)^{-t}}{\sum_{t=0}^{T=21} \text{QPW}_t * (1+r)^{-t}} \qquad (10)$$

$$\text{L.C.}^{ECY}{}_{22} = \frac{\sum_{t=0}^{T=21} C_t{}^{ECY} * (1+r)^{-t}}{\sum_{t=0}^{T=21} \text{QPW}_t * (1+r)^{-t}} \qquad (11)$$

$$\text{L.C.}^{EG}{}_{22} = \frac{\sum_{t=0}^{T=21} C_t{}^{EG} * (1+r)^{-t}}{\sum_{t=0}^{T=21} \text{QPW}_t * (1+r)^{-t}} \qquad (12)$$

**Table 1.** Table of Parameters.

| Table of Parameters | |
|---|---|
| $\text{P.V.}^{P.W.}_{t=0}$ | Financial net present value |
| r | Financial, real discount rate |
| $C_t{}^F$ | Financial cost in year *t* |
| $\text{P.M.X.}_t{}^F$ | The financial value of the payment to the municipality for raw water |
| $\text{E.C.X.}_t{}^F$ | Total financial electricity cost of pumping |
| $\text{ChCX}_t^F$ | Total financial chemicals cost |
| $\text{FO\&MX}_t^F$ | Total financial fixed O&M expenditures |
| $\text{ICAPE}_t^E$ | Total financial initial capital costs |
| $\text{RCAPEX}_t^F$ | Total financial recurrent capital costs |
| $\text{L.C.}^F$ | Total financial levelized cost of water |
| $C_t{}^E$ | The economic cost in year *t* (without pollution) |
| $C_t{}^{ENC}$ | North Cyprus economic cost in year *t* |
| $C_t{}^{ECY}$ | Cyprus economic cost in year *t* |
| $C_t{}^{EG}$ | The global economic cost in year *t* |
| $\text{PM}_t^E$ | The economic value of the payment to the municipality for raw water |
| $\text{E.C.}_t^E$ | Total economic electricity cost of pumping |
| $\text{ChC}_t^E$ | Total economic chemicals cost |
| $\text{RCAPE}_t^E$ | Total economic recurrent capital costs |
| $\text{EC}_t^{NC}$ | The economic cost of local emissions in North Cyprus |
| $\text{EC}_t^{CY}$ | The economic cost of local emissions in all of Cyprus |
| $\text{EC}_t^{GHG}$ | The economic cost of greenhouse gases |
| $\text{L.C.}^E{}_{22}$ | Total economic levelized cost of water (without pollution) 2022 prices |
| $\text{L.C.}^{ENC}{}_{22}$ | Total economic levelized cost of water (North Cyprus) 2022 prices |
| $\text{L.C.}^{ECY}{}_{22}$ | Total economic levelized cost of water (all Cyprus) 2022 prices |
| $\text{L.C.}^{EG}{}_{22}$ | Total economic levelized cost of water (global) 2022 prices |

## 5. Estimation of the Costs of Mitigating Risks of Water Shortages

### 5.1. Financial Analysis

Based on the framework outlined by equations 1 to 12 above, a detailed integrated project model has been constructed to quantify the production of water and all the costs incurred for 20 years. The values of the parameters used in the financial-economic model of this plant are presented in Appendix A.

The project model uses detailed technical as well as financial and economic parameter values to complete an integrated appraisal of the R.O. plant that uses treated wastewater as feedstock. The full Excel spreadsheet model of the integrated appraisal is available

upon request from the corresponding author. To increase the reliability of the overall water supply system, the plant mustn't be operating at full capacity all the time. There will be a need to have excess capacity much of the time so that it can respond quickly to the fluctuations of demand that occur on a daily or seasonal basis.

Using Equation (1), the quantity of water produced is estimated each year and discounted back to the initial year of operations, 2019. If the plant were to work at full capacity throughout the year, it would produce 292,000 m$^3$ per year or 5,840,000 m$^3$ over its operating life. Expressed in present value terms @ 8% as of its initial year of operation (2019), a total of 2.87 million m$^3$ of high-quality water would be produced over the plant's operating life (Table 2, row 1, column 1). However, if the plant were being used less to provide backup to offset either surge in demand or supply interruptions of other water sources, it would produce less water.

**Table 2.** Present Value of Output and Costs by Levels of Operation Capacity, 2022 Prices.

|   |   | **1** | **2** | **3** |
|---|---|---|---|---|
|   | Capacity Utilization (%) | 100 | 75 | 50 |
| 1 | Quantity of Water Produced (' 000) m$^3$ | 2867 | 2150 | 1433 |
| 2 | Total payment for wastewater (' 000) USD | 645.05 | 483.79 | 322.53 |
| 3 | Total electricity cost (' 000) USD | 1699.41 | 1281.37 | 863.33 |
| 4 | Total chemicals cost (' 000) USD | 15.54 | 11.65 | 7.77 |
| 5 | Total fixed O&M expenditures (' 000) USD | 196.02 | 198.49 | 200.97 |
| 6 | Total initial capital costs (' 000) USD | 272.87 | 272.87 | 272.87 |
| 7 | Total recurrent capital costs (' 000) USD | 280.82 | 280.82 | 280.82 |
| 8 | Total lifetime financial costs (' 000) USD | 3109.71 | 2529.00 | 1948.29 |

Given that the utilization rate of this plant on average between 50% and 75%, the levelized production and costs of water produced by the plant are also estimated for an average utilization rate of 75% and 50%. If the operation of such a plant were to average 75% or 50%, it would produce a discounted value @ 8% of 2.15 and 1.43 million m$^3$, respectively, of high-quality water over its lifetime.

By dividing the present value of each of the cost components in rows 2 to 7 in Table 2 by the present value of the water produced (Table 2, row 1), the levelized financial cost of each component of costs can be estimated for the life of the plant. These levelized financial costs (L.C.$^F_i$) are the values in Table 3 for each cost component and different average utilization levels using Equation (4). These values are expressed in USD at the price level of 2022.

**Table 3.** Levelized Financial Costs per USD/m$^3$ of Water Produced at Different Average Capacity Levels in 2022 Prices.

|   |   | **1** | **2** | **3** |
|---|---|---|---|---|
|   | Capacity utilization (%) | 100 | 75 | 50 |
| 1 | Payment to municipality | 0.225 | 0.225 | 0.225 |
| 2 | Electricity cost | 0.593 | 0.596 | 0.602 |
| 3 | Chemicals cost | 0.005 | 0.005 | 0.005 |
| 4 | **Total variable cost** | 0.823 | 0.826 | 0.833 |
| 5 | Fixed O&M cost | 0.068 | 0.092 | 0.140 |
| 6 | Initial capital cost | 0.095 | 0.127 | 0.190 |
| 7 | Recurrent capital cost | 0.098 | 0.131 | 0.196 |
| 8 | **Total fixed cost** | 0.261 | 0.350 | 0.526 |
| 9 | Levelized cost of water | 1.085 | 1.176 | 1.359 |

Full capacity is associated with twenty hours of operation per day and four hours of backwash and maintenance (Table 3, column 1). The 15- and ten h values reflect the

costs associated with the plant operating on average at 75% and 50% of its full capacity (columns 2 and 3). The levelized costs are expressed for producing 1 m$^3$ of clean water. Each component of the variable and fixed costs, as identified in Equation (2), is quantified for these four production levels.

The first three variables in Equation (2) are the variable costs of the facility. These include the payment to the municipality for the treated wastewater (P.M.X.$_{it}^F$), the power cost (E.C.X.$_{it}^F$), which comes mainly from the electricity usage of the pumps, and the chemicals cost (ChCX$_{it}^F$). In any R.O. water treatment, the electricity used to pump the water is the largest component of production costs. If the R.O. plant were to operate at full capacity, it would require approximately 1100 MWh per year. Over the project's lifetime, the undiscounted amount of electricity consumed at full capacity would be 22,000 MWh. The present value of the electricity consumed by the plant would be approximately 10,900 MWh. As the plant operates at less than full capacity, its electricity use falls almost proportionally. The long-run cost of electricity is estimated to be USD 0.17/kWh. This is equal to the real long cost of producing and delivering electricity in North Cyprus. This variable cost component ranges from a low of USD 0.59/m$^3$ when the plant is operating at full capacity to USD 0.60/m$^3$ operating at 50% of its full potential (Table 3, row 2).

These levelized variable costs for the various inputs per m$^3$ of water produced are reported in rows 1, 2, and 3 of Table 3. The estimates of total levelized variable costs are reported in Table 3, row 4. Expressed in USD, the total variable cost per m$^3$ of pure water produced ranges from a low of USD 0.82 when the plant is operating at full capacity to USD 0.83 if it is operating at 50% of its full potential (Table 3, row 4).

The final three items in Equation (2) refer to the components of the fixed costs of the R.O. plant. The first item is the fixed operating and maintenance expenses $\left(\text{FO\&MX}_{it}^F\right)$. This includes the annual cost of spare parts, administrative and accounting expenses, the operator's wages, insurance, external support, and water quality monitoring. The levelized value of these components of costs ranges from USD 0.068/m$^3$ of pure water when the plant is fully utilized to USD 0.140/m$^3$ when it is operating at 50% of its full potential (Table 3, row 5).

The sixth variable in Equation (2) is the initial capital cost of the plant. In this post-evaluation of the plant, we are considering the cost of the plant in 2019, adjusted to 2022 prices. This component includes the initial capital cost of pumps, membranes, cartridge filters, sand filters, carbon filters, storage tanks, pipelines and fittings, electricity installation, land, buildings, construction workers' wages, miscellaneous costs, and professional service costs. Expressed as USD/m$^3$ of water produced, this cost varies from USD 0.095/m$^3$ when the plant runs at full capacity to USD 0.190/m$^3$ when it can operate on average for only ten hours a day (Table 3, row 6).

The final component of fixed costs is the recurrent capital cost $\left(\text{RCAPEX}_{it}^F\right)$. That includes the periodic replacement of the pumps, membranes, and filters. In these estimations, the technical requirement is that the timing of the replacement of the pumps, membranes, and filters is not affected by the plant's capacity utilization. These recurrent costs over the plant's lifetime are as significant as the initial capital costs. If the plant were to operate at full capacity, the levelized recurrent costs would amount to USD 0.098/m$^3$ of water produced and would increase to USD 0.196/m$^3$ if the plant were to operate on average 50% of the time (Table 3, row 7).

The levelized cost of these three items of fixed costs, as reported in Table 3, row 8, amounts to USD 0.261/m$^3$ if the plant is operated at its full capacity but increases to USD 0.526/m$^3$ if it operates on average 50% of the time. These values of the fixed levelized costs are at the core of decision making in determining what degree of water service reliability is optimal for the community.

Combining levelized variable costs with levelized fixed costs, the total levelized cost of water obtained from this R.O. plant starts from USD 1.085/m$^3$ if used full-time, hence providing the minimum amount of risk mitigation. The levelized cost then increases to

USD 1.176/m$^3$ and USD 1.359/m$^3$ if it operates on average at 75% and 50% of its full capacity. An operating level of 50% of the time means that the plant is primarily used to offset the abrupt variations in demand and other sources of supply over time.

From the historical production records, this plant has been operating at between 50% and 75% of its capacity. Hence, its approximate levelized cost of water is between USD 1.359/m$^3$ and USD 1.176/m$^3$. This provides an estimate of the value private individuals in this community are willing to pay per m$^3$ of water produced to maintain stability in both the quantity and the cost of the water they need. Although the high-quality water ultimately sourced from Turkey that is now being obtained from the municipalities appears to be much less costly, its supply is associated with a high degree of uncertainty in terms of availability and price. The enhanced treatment of urban wastewater to potable quality by R.O. is how this community has managed to mitigate these risks, but at a substantial financial cost.

### 5.2. Economic Analysis

The financial analysis measures what private individuals are willing to pay for a reliable water supply system according to their revealed preference behavior. However, the economic costs may differ from the financial ones if the private financial perspective does not account for external costs and benefits. In this case, there are two important externalities. First, there are the health and damage costs of the pollution created by generating the electricity used to power the plant. Second, the price charged by the municipality for the treated wastewater used as an input into the R.O. system is far above the economic opportunity cost of this water. A third less important externality arises because when this plant uses the treated wastewater from NWTP, less water will be available to recharge the aquifer. This increases farmers' cost to pump from the aquifer because of the increased distance to the water table.

The potential environmental costs of the concentrated effluents from the R.O. plant are not included in this analysis. In this case, the environmental impact of these effluents is likely to be small or non-existent for two reasons. First, this is a relatively inefficient plant where for every 2 m$^3$ of wastewater purchased, 1 m$^3$ is returned to the source of the wastewater. Second, this plant purchases only about 5% of the treated wastewater flow from NWTP. All the rest of this plant's water goes into recharging the aquifer, along with any effluent from this R.O. plant.

### 5.3. Emission Costs

The local emissions created by using H.F.O. to power large diesel generators and two steam turbines without pollution filters result in a very high level of local pollution. The power plants are located near the prime tourist areas on the island, which are also some of the most densely populated areas. The additional costs imposed on people's health and the deterioration of property are some of the most important economic costs of this wastewater reuse.

The E.U. has estimated the monetary values of the damages incurred by residents of the whole island of Cyprus per kg of each pollutant. These values were adjusted for inflation to bring them to the price level of 2022 and expressed in USD. The estimated USD cost of the damage per kg of each pollutant is reported in Table 4, column 2.

The elements of emissions are separated into two groups. First, some emissions impact the health and physical assets of the local communities. These include sulfur dioxide, nitrogen oxides, particulate matter (2.5 μm and less in size), particulate matter (2.5 μm and less in size), and non-methane organic compounds (NMVOCs) (Table 4, rows 1 to 5). Second, there are greenhouse gases that have a global impact. These include the global emissions of carbon monoxide, carbon dioxide, and methane (Table 4, rows 6 to 8).

**Table 4.** Environmental Emission and Economic Cost of Pollutants of Diesel Fuel Generators.

| | Pollutant from Electricity Generation by H.F.O. | | Kg/MMBtu | USD/kg Emission (2022 Prices) |
|---|---|---|---|---|
| | | | 1 | 2 |
| 1 | Volatile organic compounds | NMVOC | 0.04 | −0.51 [1] |
| 2 | Nitrogen oxides | $NO_x$ | 0.86 | 8.8 [2] |
| 3 | Particulate matter | PM10 | 0.03 | 10.45 |
| 4 | Ultra-fine particulate matter | PM2.5 | 0.02 | 67.61 [3] |
| 5 | Sulfur dioxide | $SO_x$ | 0.46 | 9.94 |
| 6 | Carbon dioxide | $CO_2$ | 74.84 | 0.07 |
| 7 | Carbon monoxide | CO | 0.39 | 0.05 |
| 8 | Methane | $CH_4$ | 0.004 | 0.86 |

Note [1]: The negative value for NMVOC emissions in Cyprus is related to the fact that $NO_x$ is the primary precursor of ozone in Cyprus and that emissions of NMVOC tend to lower the ozone concentrations. Note [2] and [3]: 66.8% of Cyprus's population lives in cities, so for calculating these two values, we used 66.8% of the pollutant in urban areas and 33.2% in rural areas. Source Column 1 [40]; Source Column 2 [41].

To estimate the environmental cost of generating the electricity to power the R.O. plant, the required MWhs of electricity must be converted into the MMBtu needed by the thermal plants to generate this electricity. The values of MMBtu are then multiplied by the emission coefficients, as reported in Table 4, column 1, to obtain the kgs produced by each of the emissions. These kgs of emissions by type are then multiplied by their social costs per kg, as reported in Table 4, column 2. The present values of these emission costs over the plant's lifetime are reported in Table 5.

**Table 5.** Present value @ 8% emission costs for different levels of capacity utilization, '000 USD.

| | | 1 | 2 | 3 |
|---|---|---|---|---|
| | Capacity Utilization (%) | 100 | 75 | 50 |
| | Local emission | | | |
| 1 | Economic cost of NMVOCs | −0.70 | −0.53 | −0.36 |
| 2 | Economic cost of nitrogen oxides emission | 282.16 | 212.64 | 143.12 |
| 3 | Economic cost of particulate matter (10 μm) emission | 10.11 | 7.62 | 5.13 |
| 4 | Economic cost of particulate matter (2.5 μm) | 45.56 | 34.34 | 23.11 |
| 5 | Economic cost of sulfur dioxide emission | 169.44 | 127.69 | 85.94 |
| 6 | Sub-total of local economic cost | 506.56 | 381.75 | 256.94 |
| | Global emission | | | |
| 7 | Economic cost of carbon dioxide emission | 202.28 | 152.44 | 102.60 |
| 8 | Economic cost of carbon monoxide emission | 0.70 | 0.53 | 0.36 |
| 9 | Economic cost of methane emission | 0.12 | 0.09 | 0.06 |
| 10 | Sub-total of global cost | 203.09 | 153.06 | 103.02 |
| 11 | Economic cost of emission in North Cyprus | 253.28 | 190.87 | 128.47 |
| 12 | Economic cost of emission in all of Cyprus | 506.56 | 381.75 | 256.94 |
| 13 | Economic cost of greenhouse gases | 203.09 | 153.06 | 103.02 |
| 14 | Total economic cost emission of electricity production | 709.65 | 534.80 | 359.96 |

The present value of the total economic cost (mainly health costs) of the emissions from generating the electricity for the plant, which impacts the residents of the entire island of Cyprus, ranges from USD 506,560 if the plant is operating at full capacity to USD 256,940 if it is operating at only 50% of its capacity (Table 5, row 6). As these thermal generation plants are in the center of the most densely populated area of North Cyprus, the assumption is made that 50% of the E.U. estimates of the damage costs for the entire island would be borne by the residents of North Cyprus (Table 5, row 11).

The results show that greenhouse gas emissions will add between USD 203,090 and USD 103,020 to the social cost from the global point of view (Table 5, row 10). The E.U. recommended a social cost of carbon of USD 70/ton. Combining the local and global

emission costs, the present value of the lifetime total emission costs ranges from USD 709,650 to USD 359,960 (Table 5, row 14). When these emission costs are considered along with the present value of the lifetime financial costs of the operation, they add another 23% to the overall economic costs of the R.O. system if the plant runs at full capacity but only 18% to the financial costs of the plant if it operates at 50% of capacity.

### 5.4. Economic Opportunity Cost of Treated Wastewater

The treated wastewater is purchased from the Lefkosa Municipality Water Authority at TL1.62/m$^3$, or USD 0.10/m$^3$. Over the past four years, this has been approximately the prevailing USD price charged for the wastewater. This also approximates the EUR0.15/m$^3$ that farmers in South Cyprus pay in 2022 for treated wastewater [42]. If this facility did not purchase the treated wastewater, it would be left to recharge the aquifer. The ultimate economic benefit would be raising the aquifer's level and reducing the pumping costs of farmers in the region who are pumping water from wells. It is estimated that the savings in pumping costs from having more water in the aquifer is approximately USD 0.01/m$^3$ [30]. This cost is based on the amount of water produced, not the amount purchased. Due to the inefficiency of this plant, approximately half of the water purchased is returned to the wastewater source to soak into the aquifer. For this lifetime economic cost of the wastewater, input is the economic cost per m$^3$ (USD 0.01) multiplied by the quantity of final water output.

### 5.5. Results of Economic Analysis

The economic costs ($C_{it}{}^E$) of the water produced by the plant, excluding and including the cost of pollution emissions borne by the specific parties, are defined by Equations (5–8). The present values of each economic cost, including both the resource costs of production and the externalities, are reported in Table 6.

**Table 6.** Present value @ 8% Total Economic Costs for Different Levels of Capacity Utilization, '000 USD.

|  |  | 1 | 2 | 3 |
|---|---|---|---|---|
|  | Capacity Utilization (%) | 100 | 75 | 50 |
| 1 | Total economic opportunity cost of wastewater | 28.67 | 21.50 | 14.33 |
| 2 | Total economic cost of electricity | 1699.41 | 1281.37 | 863.33 |
| 3 | Total economic cost of chemicals | 15.54 | 11.65 | 7.77 |
| 4 | **Total variable cost** | 1743.61 | 1314.52 | 885.43 |
| 5 | Total economic cost of initial capital | 272.87 | 272.87 | 272.87 |
| 6 | Total economic cost of recurrent capital cost | 280.82 | 280.82 | 280.82 |
| 7 | Total economic cost of fixed O&M | 196.02 | 198.49 | 200.97 |
| 8 | **Total fixed cost** | 749.71 | 752.19 | 754.66 |
| 9 | Total North Cyprus emission cost | 253.28 | 190.88 | 128.47 |
| 10 | Total all Cyprus emission cost | 506.56 | 381.75 | 256.94 |
| 11 | Total cost greenhouse gas emission | 203.10 | 153.06 | 103.02 |
| 12 | Total economic cost of water (without pollution) | 2493.32 | 2066.71 | 1640.09 |
| 13 | Total economic cost of water (North Cyprus emission) | 2746.60 | 2257.58 | 1768.56 |
| 14 | Total economic cost of water (all Cyprus emission) | 2999.88 | 2448.46 | 1897.03 |
| 15 | Total economic cost of water (local and global emission) | 3202.98 | 2601.51 | 2000.05 |

These present values of economic costs, when divided by the present values of the water produced (Table 2, row 1), yield the levelized economic costs of producing water by this R.O. plant from the perspective of the residents of North Cyprus, the whole island of Cyprus, and globally. The levelized economic costs is estimated using these values and Equations (9–11), respectively. The results are reported in Table 7. The first three variables in these equations are the economic variable costs of the facility, including the economic value of the municipality's treated wastewater ($PM_{it}^E$) (row 1), the economic electricity cost

(E.C.$_{it}^{E}$) (row 2), and the economic chemicals cost (ChC$_{it}^{E}$) (row 3). These costs are expressed in their levelized costs per m$^3$ of water produced. The next three items in the equations (rows 5, 6, and 7 in Table 7) refer to the levelized economic fixed costs of the R.O. plant.

**Table 7.** Levelized Economic Costs in USD/m3 for Different Levels of Utilization (2022 Prices).

|    |                                                          | 1     | 2     | 3     |
|----|----------------------------------------------------------|-------|-------|-------|
|    | Capacity Utilization (%)                                 | 100   | 75    | 50    |
| 1  | Levelized economic opportunity cost of wastewater        | 0.010 | 0.010 | 0.010 |
| 2  | Levelized economic cost of electricity                   | 0.593 | 0.596 | 0.602 |
| 3  | Levelized economic cost of chemicals cost                | 0.005 | 0.005 | 0.005 |
| 4  | **Total levelized variable cost**                        | 0.608 | 0.611 | 0.618 |
| 5  | Levelized economic cost of initial capital               | 0.095 | 0.127 | 0.190 |
| 6  | Levelized economic cost of recurrent capital             | 0.098 | 0.131 | 0.196 |
| 7  | Levelized economic cost of fixed O&M                     | 0.068 | 0.092 | 0.140 |
| 8  | **Total levelized fixed cost**                           | 0.262 | 0.350 | 0.526 |
| 9  | Levelized cost of North Cyprus emission                  | 0.088 | 0.089 | 0.090 |
| 10 | Levelized cost of all Cyprus emission                    | 0.177 | 0.178 | 0.179 |
| 11 | Levelized cost of greenhouse emission                    | 0.071 | 0.071 | 0.072 |
| 12 | Levelized economic cost of water (without pollution)     | 0.870 | 0.961 | 1.144 |
| 13 | Levelized economic cost of water (North Cyprus emission) | 0.953 | 1.050 | 1.234 |
| 14 | Levelized economic cost of water (all Cyprus emission)   | 1.042 | 1.139 | 1.323 |
| 15 | Levelized economic cost of water (global emission)       | 1.117 | 1.210 | 1.395 |

The levelized economic costs (L.C.$^{E}_{i}$) of the current plant are summarized in Table 7 from four different perspectives. First, the levelized costs of the production from the R.O. plant are estimated by excluding any pollution cost (L.C.$^{E}_{i}$) from electricity generation (Table 7, row 12). Second, the levelized costs are estimated from the perspective of North Cyprus residents (L.C.$^{ENC}_{i}$) (Table 7, row 13). In this case, none of the costs of the damage from greenhouse gases are included, but half of the island-wide costs of the local emissions impacts are included. Third, the levelized economic costs are estimated for all residents of the island of Cyprus (L.C.$^{ECY}_{i}$) (Table 7, row 14). In this case, the entire estimated costs of the local emissions are included, while all the economic costs of the additional greenhouse costs are excluded. Finally, the levelized costs from the global perspective (L.C.$^{EG}_{i}$) are reported in Table 7, row 15. In this case, the emission costs that have a local impact are added to the economic cost of the damage from additional greenhouse gas costs.

In this last case, where all emissions per m$^3$ are taken into consideration, the levelized economic costs of water production will vary from USD 1.117/m$^3$ if used at full capacity to USD 1.210/m$^3$ and USD 1.395/m$^3$, respectively, if utilization falls to 75% and 50% of capacity (Table 7, row 15). Of course, the utilization level must be below 100% if this water source is available to fill the gaps between the demand for and supply of water when they arise. These values reflect the global cost of maintaining water reliability in North Cyprus. Of these costs, 22% to 18%, 18–22% of the total economic costs depending on the capacity level, arising from the pollution created by the electricity system that provides power to the R.O. system.

If none of the emission costs are included, the levelized economic cost for the plant operating at full, 75%, and 50% capacity varies across these three categories from USD 0.870/m$^3$ to USD 1.144/m$^3$ (row 12). When just the local emission costs borne by North Cyprus residents are included, the levelized economic costs increase to between USD 0.953/m$^3$ and USD 1.234/m$^3$ (row 13). These are the levelized economic costs of water borne by the economy of North Cyprus as the operator attempts to use this facility to maintain an overall reliable water supply.

When all the island of Cyprus residents are included in the economic costs, the levelized economic cost increases to between USD 1.042/m$^3$ and USD 1.323/m$^3$ (row 14).

Comparing the levelized economic costs of water produced by this plant (Table 7) with its levelized financial cost (Table 3), it is surprising how close these two sets of cost estimates are. In all cases, the economic cost of producing potable water with this R.O. plant is less than USD 0.03/m$^3$ more expensive than its levelized financial cost. At the same time, very substantial emission costs add to the economic cost of operating this plant and are not part of the levelized financial costs. A distributive analysis must be conducted to reconcile these two sets of costs.

## 6. Evaluation of Distributive Impacts on Water Risk Mitigation

The fundamental rule of a distributive analysis is that the economic present value of a set of variables is equal to the financial present value of these variables plus the present value of the sum of the externalities of the project. The values of these externalities are the distributive impacts [33]. This relationship is expressed in Equation (13).

$$PV_{t=0}^{economic} = PV_{t=0}^{financial} + \sum_i PV_{t=0,\,i}^{externalities} \tag{13}$$

Expressed in another way is the difference between the present value of the economic values of a set of variables and the present value of their financial values equal the sum of the present values of the externalities. In this case, the externalities are the emission costs inflicted on the residents of North Cyprus, the emission costs inflicted on the residents of South Cyprus, the cost the GHG emission on the global residents, the higher pumping costs imposed on the farmers extracting water from the local aquifer and the benefits received by the local municipality from the sale of the wastewater to the plant.

The present value of the economic costs of this plant is obtained from Table 6, row 15, and reported in Table 8, row 1, for different utilization levels. In this case, the values reflect the total global costs of investing and operating the plant, including local and global pollution costs. The present value of the lifetime financial costs of the plant is reported in Table 8, row 2. The differences between the economic and the financial present values of costs are reported in Table 8, row 3. These values represent the present value of the sum of the impacts on all the different parties. While the net differences between the present values of all the combined externalities appear to be relatively small, amounting to 2% and 3% of total financial costs, this net difference hides some significant impacts.

**Table 8.** Distributive Impacts of Different Levels of Utilization in 2022 Prices, '000 USD.

|  |  | **1** | **2** | **3** |
|---|---|---|---|---|
|  | Capacity utilization (%) | 100 | 75 | 50 |
| 1 | P.V. of economic costs | 3202.98 | 2601.51 | 2000.05 |
| 2 | P.V. of financial costs | 3109.71 | 2529.00 | 1948.29 |
| 3 | P.V. of net distributive impacts (rows 1–2) | 93.27 | 72.51 | 51.76 |
|  | Parties impacted by externalities |  |  |  |
| 4 | Economic cost borne by residents of North Cyprus | 253.28 | 190.87 | 128.47 |
| 5 | Economic cost borne by residents of South Cyprus | 253.28 | 190.87 | 128.47 |
| 6 | The economic cost of greenhouse gases | 203.09 | 153.06 | 103.02 |
| 7 | P.V. of additional pumping costs imposed on farmers | 28.67 | 21.50 | 14.33 |
| 8 | P.V. of payments to the municipality for wastewater | 645.05 | 483.79 | 322.53 |
| 9 | PV of net distributive impacts (rows 4 + 5 + 6 + 7 − 8) | 93.27 | 72.51 | 51.76 |

The disaggregation of the distributive impacts is reported in Table 8, rows 4 to 8. In row 8, the plant owner's payments to the local municipality are reported. These payments are financial but not economic costs as they are transfers to the local municipality. This can be viewed as simply a tax of between USD 645,050 and USD 322,530 to be paid over the life of this project. Expressed as a percentage of the total economic costs, these charges are equal to between 20% and 16% of the present value of the total lifetime financial cost of

the plant. This is a substantial positive benefit to the local municipality arising from the operation of this R.O. plant.

The opportunity cost of the wastewater used in the plant is reported in Table 8, row 7. The present value of this variable ranges from USD 28,670 to USD 14,330. This is an economic resource cost as it reflects the higher pumping costs that farmers will incur because the water table level is reduced. It is the farmers who will bear this cost.

The local environmental damage caused by the pollution created by electricity generation is reported in Table 8, row 4, as the amount of costs borne by the residents of North Cyprus. The environmental costs inflicted on the residents of South Cyprus are reported in row 5. It is assumed that the total local environmental costs are split 50/50 between North and South Cyprus residents. These costs range from USD 253,280 to USD 128,470 for each of the Cypriot communities, depending on the level of plant utilization. These costs are very substantial. The environmental costs range from 8% to 4% of the plant's total economic costs.

The final externality arises from the increase in greenhouse gas emissions produced by thermal electricity generation plants. These costs are evaluated using E.U. recommended values for the social cost of carbon emissions, which range from USD 203,090 to USD 103,020. The percentage of costs ranges from 6% to 3 % of the total lifetime economic costs.

## 7. Conclusions

This analysis shows that people living in arid conditions highly value water service reliability. Furthermore, it demonstrates how enhanced wastewater treatment through an R.O. process is an effective way to introduce it into a conventional water supply system to manage the risks of water shortages. With a modest amount of water storage, the R.O. system using wastewater as its feedstock can adjust quickly so that every gap between the demand for water and conventional supplies can be met almost instantaneously.

As has been extensively discussed in the literature, given the scale of the plant, the cost of electricity is usually the most critical determinant of water production's financial costs by R.O. In the case of wastewater reuse, this analysis shows that the price that the system must pay for the input of treated wastewater in combination with the efficiency of the R.O. system is also an essential determinant of the financial cost of producing potable water in this way.

Except for the impact on farmers, each externality's impact is substantial. However, given the very significant charges being paid for the treated wastewater by the plant owners, they fully compensate the residents of North Cyprus more than twice the amount of the environmental costs they bear due to the increased electricity generation.

In analyzing the economic cost of wastewater reuse and purification by R.O., a critical factor is how the electricity that will power the system is generated. In this case, using H.F.O. as fuel in these large diesel plants imposes a high cost on the environment and the health of the residents of Cyprus.

**Author Contributions:** Conceptualization, G.P.J. and F.N.C.; methodology, G.P.J.; formal analysis, F.N.C.; resources, M.H.; data curation, M.H.; writing—original draft preparation, F.N.C.; writing—review and editing, G.P.J.; supervision, H.J. All authors have read and agreed to the published version of the manuscript.

**Funding:** This research received no external funding.

**Institutional Review Board Statement:** The study did not require ethical approval.

**Informed Consent Statement:** Not applicable.

**Data Availability Statement:** In Appendix A, we provided the data.

**Acknowledgments:** The authors wish to thank Mete Boyaci, Mostafa Alkaravli, and Cafer Aytac of the Levent Group Ltd. for providing data and their time to explain the role and operation of this water system. Serkan Abbasoglu and Arash Peykanfer provided critical assistance in informing us of the nature and environmental impacts of the electricity system in North Cyprus. A special thanks to

Mehrshad Radmehr and the members of the Center for Applied Research in Business, Economics, and Technology (CARBET) at Cyprus International University for organizing two seminars in which the ideas in this paper could be presented, and many valuable suggestions were received.

**Conflicts of Interest:** The authors declare no conflict of interest.

## Appendix A

**Table A1.** Parameter Values Used in Modeling of Levent Wastewater RO Plant.

| Timing Assumptions | | |
|---|---|---|
| Construction duration | 1 | Year |
| Operations duration | 20 | Year |
| Liquidation year | 21 | Year |
| Replacement of carbon filters | Every 3 years | |
| Replacement of membranes | Every 3 years | |
| Pump replacement | Every 10 years | |
| Cartridge filter replacement | Every 2 months | |
| Water characteristics | | |
| Flow rate | 120 m$^3$/hr | |
| Input capacity per hour (actual flow rate due to friction) | 90 m$^3$/hr | |
| Output of potable water per hour operating 20 hr/day | 40 m$^3$/hr | |
| Operating R.O. | 1–20 h | |
| Backwash R.O. | 4 h | |
| 1. Capital expenditure (in USD (2021 prices)) | | |
| Equipment | | |
| Pumps cost | | |
| Wastewater intake pumps installation | | |
| Number of wastewater intake pumps in operation | 1 | # |
| Number of working hours per day | 1–20 | Hours |
| Cost per wastewater intake pump | 1905.49 | USD |
| Pump to transfer from wastewater storage to R.O. system | | |
| Number of pumps | 1 | # |
| Number of working hours per day | 1–20 | Hours |
| Cost of pump | 1905.49 | USD |
| High-pressure pumps installation | | |
| Number of high-pressure pumps in operation | 2 | # |
| Number of working hours per day | 1–20 | Hours |
| Cost per high-pressure pump | 8989.00 | USD |
| Membrane backwash pumps installation | | |
| Number of membrane backwash pumps | 1 | # |
| Number of working hours per day | 4 | Hours |
| Cost of membrane backwash pumps | 512.00 | USD |

**Table A1.** *Cont.*

| Timing Assumptions | | |
|---|:---:|:---:|
| Product transfer pumps installation | | |
| Number of product transfer pump in operation | 1 | # |
| Number of working hours per day | 1–20 | Hours |
| Cost per transfer pump | 1905.49 | USD |
| RO membranes installation | | |
| Number of R.O. membranes | 60 | # |
| Cost per R.O. membrane | 1000.00 | USD |
| Filters | | |
| Cartridge filters installation | | |
| Number of filters | 3 | # |
| Cost per filter | 200.00 | USD |
| Sand filters installation | | |
| Number of filters | 2 | # |
| Cost per filter | 11,425.00 | USD |
| Carbon filters installation | | |
| Number of filters | 2 | # |
| Cost per filter | 13,360.00 | USD |
| Storage tank | | |
| Storage tank (250 m$^3$) | 2 | # |
| Cost per tank | 18,000.00 | USD |
| Buildings, plant, and equipment | | |
| Pipelines and fittings | 2250.00 | USD |
| Cost of electricity installations | 5000.00 | USD |
| Land (15,000 m$^2$) | 50,000.00 | USD |
| Containment buildings (200 m$^2$ at USD 250/m$^2$) | 50,000.00 | USD |
| Professional services (% of capital cost) | | |
| Design and consultancy | 0.05 | % |
| Miscellaneous costs | | |
| Miscellaneous cost (% of plant cost) | 0.05 | % |
| 2. Operating & maintenance expenditure | | |
| Payment to municipality for raw water per m$^3$ | 0.10 | USD/m$^3$ |
| Chemicals | | |
| Scale inhibitor per m$^3$ potable water for one hour operation | 0.01 | L/m$^3$ |
| Cost per liter of scale inhibitor | 7.02 | USD/L |
| Ferric chloride per m$^3$ potable water for one hour operation | 0.01 | L/m$^3$ |
| Cost per liter of ferric chloride | 32.00 | USD/L |
| Electricity consumption of each pump | | |
| Wastewater intake pump | 30.00 | KWh/hr |
| Wastewater storage to R.O. system | 15.00 | KWh/hr |
| High-pressure pump | 45.00 | KWh/hr |
| Product transfer pump | 15.00 | KWh/hr |

**Table A1.** *Cont.*

| Timing Assumptions | | |
|---|---|---|
| Membrane backwash pumps | 11.00 | KWh/hr |
| Total electricity consumption | 161.00 | KWh/hr |
| Cost of electrical energy | | |
| Cost/KWh (year 0 prices) | 0.17 | USD/KWh |
| Annual spare parts cost (% of CAPEX) | 0.01 | % |
| Administration and accounting | | |
| Number of administration and accounting | 0.50 | # |
| Payment per person per month | 1500.00 | USD/Month |
| Operators | | |
| Number of operators | 1 | # |
| Payment per operator | 576.00 | USD/Month |
| Insurance | 1000.00 | USD/Year |
| External support (engineering faculty) | 50.00 | USD/Month |
| Water quality monitoring | 50.00 | USD/Month |
| 3. Tariff structure | | |
| Overall water price TL | 8.40 | TL/m$^3$ |
| Overall water price USD | 0.64 | USD/m$^3$ |
| Economic value of water produced | 1.00 | USD/m$^3$ |
| 4. Working capital | | |
| Accounts receivable (% of sales revenues) | 0 | % |
| Accounts payable (% of electricity costs) | 8 | % |
| Desired cash balance (% of labor costs) | 1 | % |
| 5. Taxes and exchange rates | | |
| Real exchange rate | | |
| USD exchange rate Turkish lira USD/TL | 13.12 | |
| USD exchange rate euro USD/EUR | 0.89 | |
| Average euro exchange rate in 2022 to USD EUR/USD | 1.08 | |
| Cumulative price change E.U.R. 2015 to E.U.R. 2022 | 18.11 | % |
| Discount rates | | |
| Financial real discount rate | 8 | % |
| National parameters | | |
| Economic opportunity cost of capital (EOCK) | 8 | % |
| Foreign exchange premium (F.E.P.) | 0 | % |

Source: All the data used to carry out this analysis was obtained from the operating records of the Levent Group, the owner and operator of this plant.

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
