# Peer review of "Wastewater Reuse to Mitigate the Risk of Water Shortages: An Integrated Investment Appraisal"

_water, doi:10.3390/w14233859_

Round 1

Reviewer 1 Report

I am not an economist and thus had a hard time following the methodology and conclusions. Nevertheless, I do believe that the economic point of view is crucial. 

That said, the introduction is poorly written and a language editor is highly recommended.  Examples of such can be found below:

Introduction

L27 “have faced increasing challenges in meeting their growing water demand in urban areas” is the past tense correct? Are these challenges over?

L30 “due to the prospect of wider temperature fluctuations” Where does this come from? And what do “temperature fluctuations” has to do with water demand? Why not precipitation fluctuations?

L32 “Planning and management issues in the water sector are evolving” Rephrase

The comparison to the management of electricity supply is interesting but poorly written and very few details are given as to how electricity production is managed. Also, no reference is given to the management of electricity although this subject is well discussed in the literature. Add description and reference(s)!

L48-49 “In this paper, a similar approach is taken to assess how to maintain water service reliability by building reverse osmosis (RO) plants that utilize urban wastewater” It is not clear – similar to what? If you explain the approach to electricity supply better it will be much clearer.

L49… “Such water supplies can be tapped quickly, are completely reliable, and provide high-quality water at a cost that is not exorbitant when the gap between daily demand and the available supply changes unpredictably.” Saying that wastewater-RO systems are “completely reliable” seem like an overshoot…

General comment: The use of "effective", "completely reliable" and such should be avoided - the MS is doing an economic value and should stick to such.

In general, I think the MS has many merits but the way it is currently written its effectiveness is limited. Writing parts of the results in a more generalistic language might help non-economist understand the bottom lines better

Author Response

Dear Reviewer,

Thank you for the opportunity to revise and resubmit the manuscript. We appreciate the feedback on our manuscript “Wastewater Reuse to Mitigate the Risk of Water Shortages: A Stakeholder Analysis.” The comments were helpful and encouraged us to revise the manuscript substantially. We are much happier with the final product and appreciate the opportunity to refine and improve the manuscript. We have included the reviewers’ comments in the attached word file and provided our responses to them. In addition, all changes based on the reviewers’ suggestions have been highlighted in RED color in the revised version.

Please let us know if there is anything else you would like us to do.

Reviewer 2 Report

Contrary to what is claimed, a stakeholder analysis is by no means the focus of the paper. Stakeholder analyses today are usually conducted in a socio-empirical manner. Rather, based on the approach of integrated capital budgeting, the financial and economic costs for the investment and operation of the plant are quantified and supplemented by a schematic consideration according to so-called stakeholder impacts; however, these are in no way empirically validated, so that the term stakeholder analysis is misleading (I would therefore suggest modifying the title of the paper, even if this is how the term stakeholder analysis was used 20 years ago).

The method itself is not presented in a literature-based manner. In this respect, chapter 4 needs to be revised strongly. The announced paper by Jenkins, Juo, and Harberger is missing from the bibliography (instead, a paper by Jenkins from 1999 is given).

The literature review does not use papers published during the last years such as Marin, P., Charalambous, B., & Davy, T. (2018). Securing Potable Water Supply under Extreme Scarcity: Lessons and Perspectives from the Republic of Cyprus. World Bank; Pluchinotta, I., Giordano, R., Zikos, D., Krueger, T., & Tsoukiàs, A. (2020). Integrating problem structuring methods and concept-knowledge theory for an advanced policy design: lessons from a case study in Cyprus. Journal of Comparative Policy Analysis: Research and Practice, 22(6), 626-647 or GökçekuÅŸ, H., Kassem, Y., Quoigoah, M. P., & Aruni, P. N. (2023). Climate change, water resources, and wastewater reuse in Cyprus. Future Technology, 2(1), 1-12.

Current expertise shows that there are different ways of processing the water for different uses (cf Nahrstedt et al. Reuse of municipal wastewater for different purposes based on a modular treatment concept. J. Water-reuse 2020). The European Union has a new legislation not discussed in the paper (cf. Truchado et al. (2021). New standards at European Union level on water reuse for agricultural irrigation: Are the Spanish wastewater treatment plants ready to produce and distribute reclaimed water within the minimum quality requirements? International Journal of Food Microbiology, 356, 109352.) If there is in Cyprus a need for an RO-based water supply system with high reliability, the research question therefore is not what is the cost of maintaining the entire system, but what is the water quality(s) required to maintain the reliability of the entire water supply system, and what are the costs incurred in the various ways to achieve it? It is rather inadequate to state that the WWTP "produceces water above the European standards for treated wastewater" and that the RO produces "near-potable water". From what chemical and microbiological load to what is the water treated, for what purpose?

"While the concept is sound, there is a need to operationalize these principles in urban water system planning. One contribution of the current study is to show how these concepts can be applied in practice." Unfortunately, I missed the corresponding execution. It is not discussed it does to what extent which form of water reuse contributes to avoiding the risks of water shortages in Cyprus. The considerations of replacing electricity production with heavy oil by natural gas are not really well-founded in view of the current energy crisis in Europe. Why are regenerative forms of energy generation such as wind or sun missing here?

Author Response

(The authors gave the same response as above.)

Round 2

Reviewer 2 Report

Thank you for taking the suggestions of my review.